# Focal neural perturbations reshape low-dimensional trajectories of brain activity supporting cognitive performance

Kartik K. Iyer 🕩 [1]✉, Kai Hwang[2,3], Luke J. Hearne[1], Eli Muller[4], Mark D'Esposito[3], James M. Shine 🕩 [4] & Luca Cocchi 🕩 [1]✉

The emergence of distributed patterns of neural activity supporting brain functions and behavior can be understood by study of the brain's low-dimensional topology. Functional neuroimaging demonstrates that brain activity linked to adaptive behavior is constrained to low-dimensional manifolds. In human participants, we tested whether these low-dimensional constraints preserve working memory performance following local neuronal perturbations. We combined multi-session functional magnetic resonance imaging, non-invasive transcranial magnetic stimulation (TMS), and methods translated from the fields of complex systems and computational biology to assess the functional link between changes in local neural activity and the reshaping of task-related low dimensional trajectories of brain activity. We show that specific reconfigurations of low-dimensional trajectories of brain activity sustain effective working memory performance following TMS manipulation of local activity on, but not off, the space traversed by these trajectories. We highlight an association between the multi-scale changes in brain activity underpinning cognitive function.

[1] QIMR Berghofer Medical Research Institute, Brisbane, QLD 4006, Australia. [2] Department of Psychological and Brain Sciences and The Iowa Neuroscience Institute, The University of Iowa, Iowa City, IA 52242, USA. [3] Helen Wills Neuroscience Institute, University of California, Berkeley, CA 94720-1650, USA. [4] The University of Sydney, Sydney, NSW 2050, Australia. ✉email: Kartik.Iyer@qimrberghofer.edu.au; Luca.Cocchi@qimrberghofer.edu.au

C ognitive functions are supported by the activity of large-scale brain networks that adhere to a relatively low-dimensional manifold[1,2]. The study of low-dimensional brain dynamics allows us to capture whole-brain functional interactions between remote neural populations underpinning behavior[3]. Accordingly, recent functional neuroimaging work has shown that trajectories of brain activity evolving on low-dimensional manifolds scale with cognitive task complexity and are sensitive to performance errors[4]. These trajectories reflect how a system's activity changes according to the velocity vector field in state-space[5,6]. In essence, the vector field defines the rules of the system's behavior according to the constraints of a specific task and inform the evolution of trajectories of brain activity onto a low-dimensional sub-space (manifold)[5,7]. The term "manifold" implies that the system's dynamics can be relatively low-dimensional and smooth. Thus, changes in cognitive demands may alter the vector field and the trajectories of brain activity occupying a sub-space in state-space[2,7–9].

Findings from previous neuroimaging work suggest that a reshaping of the trajectories of brain activity evolving on low-dimensional manifolds may capture key functional adaptations to offset perturbations of neural activity in specialized brain regions supporting cognitive functions[1,4]. If, and to what degree, the reshaping of low-dimensional trajectories is causally affected by local neural perturbations remains unknown. Addressing this question is critical to bridge the knowledge gap on the causal relation between local and system-wide brain activity patterns supporting cognitive functions[10].

We addressed this knowledge gap by combining within-subject multi-session neuroimaging (functional MRI, fMRI), a validated working memory task (n-back), a non-invasive brain stimulation approach known to induce changes to local neural activity (transcranial magnetic stimulation, TMS[11]), and methods translated from the fields of complex systems and computational biology[12]. Healthy participants performed the n-back task, comprising two levels of difficulty (recalling picture items one or two items back), while in the MRI scanner[13]. Following this baseline fMRI session, targeted perturbations of spontaneous neural activity were induced on a task-relevant brain region (intraparietal sulcus, iPS; "Methods") or a task-irrelevant region (primary somatosensory cortex, S1). The iPS and the S1 TMS sessions were counterbalanced across participants. Immediately following TMS (continuous theta-burst stimulation, cTBS[11]), participants performed the same task in the scanner again ("Methods"). We defined session (baseline, S1, iPS), task (low versus high cognitive load), and behaviorally specific (correct versus incorrect performances) low-dimensional trajectories using data reduction and a state-of-the-art embedding technique (potential of heat diffusion for affinity-based transition embedding method, PHATE[12]). Here, we show that targeted perturbations of neural activity on a task-relevant brain region (iPS) rather than a region not engaged by the task (S1) caused a specific reshaping of low-dimensional trajectories of brain activity that relates to changes in task performance.

## Results
We began by defining the low-dimensional trajectories of brain activity associated with the working memory task, across the three experimental sessions ("Methods"). We estimated trial-locked fMRI responses by using a mixture of basis functions that capture both the amplitude and the temporal characteristics of the hemodynamic response ("Methods"). Next, we projected the estimated responses from low (1-back) and high (2-back) working memory load trials into a low-dimensional embedding space and calculated the trajectories of brain activity from the top three

dimensions (Fig. 1a). In line with previous work[1,4], our initial analyses showed that the session-specific trajectories were primarily captured by the first three dimensions (PHATE1, PHATE2, PHATE3) (Supplementary Table 1 and Supplementary Fig. 1). These low-dimensional trajectories were estimated using a recently developed embedding technique (PHATE), which tracks both local and global nonlinear structures in the data, offering superior state-space embedding over standard dimensional reduction techniques like t-SNE and UMAP[12].

**Effects of session and cognitive load on low-dimensional brain dynamics**. By adopting meta-analytic data from the NeuroSynth repository[14], we confirmed that our low-dimensional embeddings mapped onto higher order spatial maps linked to cognitive functions including working memory and numerical cognition (Supplementary Fig. 2). Additional analysis provided preliminary support for our initial prediction, wherein TMS-induced perturbation on iPS caused an increase in fMRI signal in PHATE1 (sessions by PHATE dimensions interaction, repeated measures ANOVA: $F_{(4,144)} = 8$, $p = 7.5 \times 10^{-6}$; see Supplementary Fig. 3a for further post-hoc test results) but a decrease in the task-related variance explained by this dimension (Supplementary Table 1). The variance captured by PHATE dimensions identifies the dominant low-dimensional signal in state-space, reflecting the global dynamics of brain activity present across experimental sessions and task conditions. Our findings suggest that PHATE1 plays a key role in supporting task-related low-dimensional activity (83% average variance across load conditions, for baseline and S1; 72% for iPS). The detected shift in task-related variance towards PHATE2 following perturbation of iPS (Supplementary Table 1) is likely to reflect an adaptive response in the system's brain dynamics. As reported in our previous work[13], these differences cannot be explained by group-level changes in behavioral performances (accuracy or reaction time) across the three experimental sessions (baseline, S1, and iPS).

Using the first three PHATE dimensions, we derived within-session group-level embedding spaces across successful ("correct") 1-back and 2-back working memory trials (Fig. 1b). Within these group embedding spaces, low-dimensional trajectories between 1-back and 2-back trials expanded (the overall length of the trajectory increased): ~1.3× for the baseline session and ~1.5× for the S1 session. For the iPS session, the expansion of low-dimensional trajectories of activity as a function of increased working memory load was more prominent and statistically significant (~7.3×, $p_{FDR} = 3 \times 10^{-16}$, iPS correct 2-back > iPS correct 1-back) (Fig. 1b). At the scale of interacting brain network communities (Fig. 1b), the iPS-related low-dimensional expansion mapped onto increased involvement of brain regions in the task-relevant associative network community (Supplementary Fig. 3b). To further quantify this low-dimensional expansion, we examined the difference (total summed Euclidean distance) between the within-session group trajectory and the trajectories of each participant. This analysis revealed a linear scaling of trajectories as a function of increased working memory load for participants during the baseline and the S1 experimental sessions (Fig. 1c). This scaling was absent in the iPS session (Fig. 1c), suggesting that TMS-induced perturbation of iPS caused a change in how the system accommodates increased cognitive load.

To further examine the functional significance of the low-dimensional reconfigurations as a function of increased working memory load, we investigated differences between correct and incorrect trials in a shared embedding space. The ability to successfully upscale working memory was linked to a generalized expansion of the manifolds' trajectories across participants (all sessions collapsed: 2.6×, $p_{FDR} = 1.2 \times 10^{-5}$, Fig. 2a). This

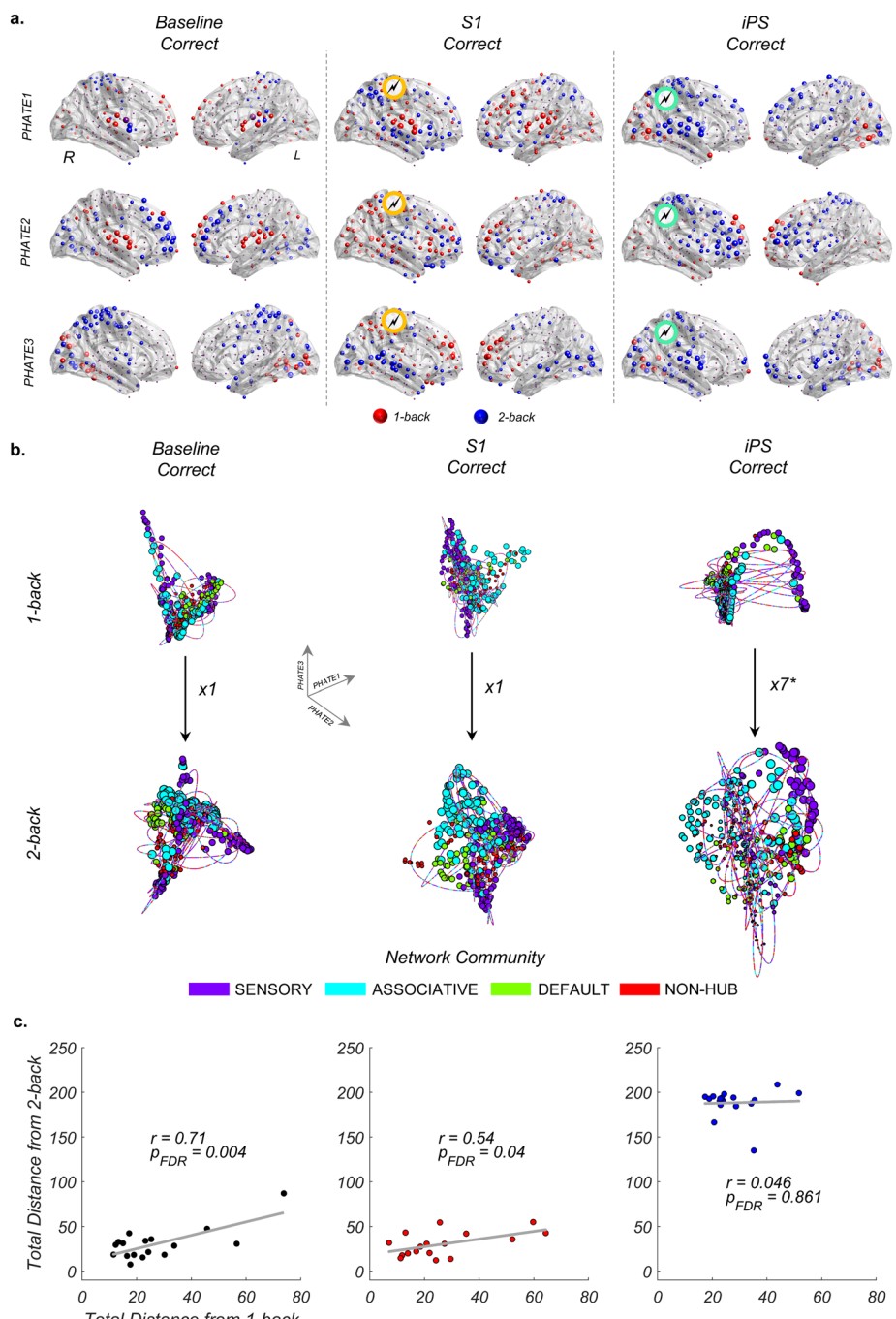

**Fig. 1 Selective impact of local brain stimulation on the low-dimensional trajectories of brain activity across working memory loads and experimental sessions. a** Spatial maps for the top three PHATE dimensions during n-back working memory tasks (1-back = red, 2-back = blue; the size of the spheres represents the relative weight of PHATE values; see "Methods"). Network communities adopted from Gordon et al.[15] ("Methods"). The majority of nodes load onto task-positive brain regions constituting the visual, cingulo-opercular, dorsal attention, and frontoparietal networks. **b** The state-space embedding of trajectories of brain activity across working memory loads during 'correct' responses across baseline, S1 (task-irrelevant region targeted by TMS, indicated as yellow in **a**), and iPS (task-relevant region targeted via TMS, indicated as green in **a**). The size of the nodes represent the relative weight of PHATE values. The color of the trajectories indicates the engagement of brain regions belonging to specific network communities. Here, asterisks indicate significant values (*$p_{FDR}$ < 0.001). **c** For each session, we calculated the difference between the within-session group trajectory and each participant's trajectory (total summed Euclidean distance) during 1-back and 2-back trials. The relationship between these load-related trajectory distances revealed a load-induced scaling in the baseline and the S1 sessions, but not in the iPS session. Source data are provided as a Source data file.

expansion was noticeable for correct 2-back trials (correct versus incorrect: 3.4×, $p_{FDR}$ = 1.2 × 10^{-5}, Fig. 2a) and was most prominent for 2-back correct trials in the iPS session (8.1×, $p_{FDR}$ = 1.8 × 10^{-16}, Fig. 2b) compared to correct trials in the baseline and the S1 sessions (expansions for baseline and

S1 sessions showed in Supplementary Fig. 4). Accordingly, correct responses in the most challenging working memory condition following iPS stimulation relied on a larger expansion of the low-dimensional trajectory compared to responses in the baseline and the S1 sessions (Fig. 2c, d, $F_{(2,48)}$ = 229,

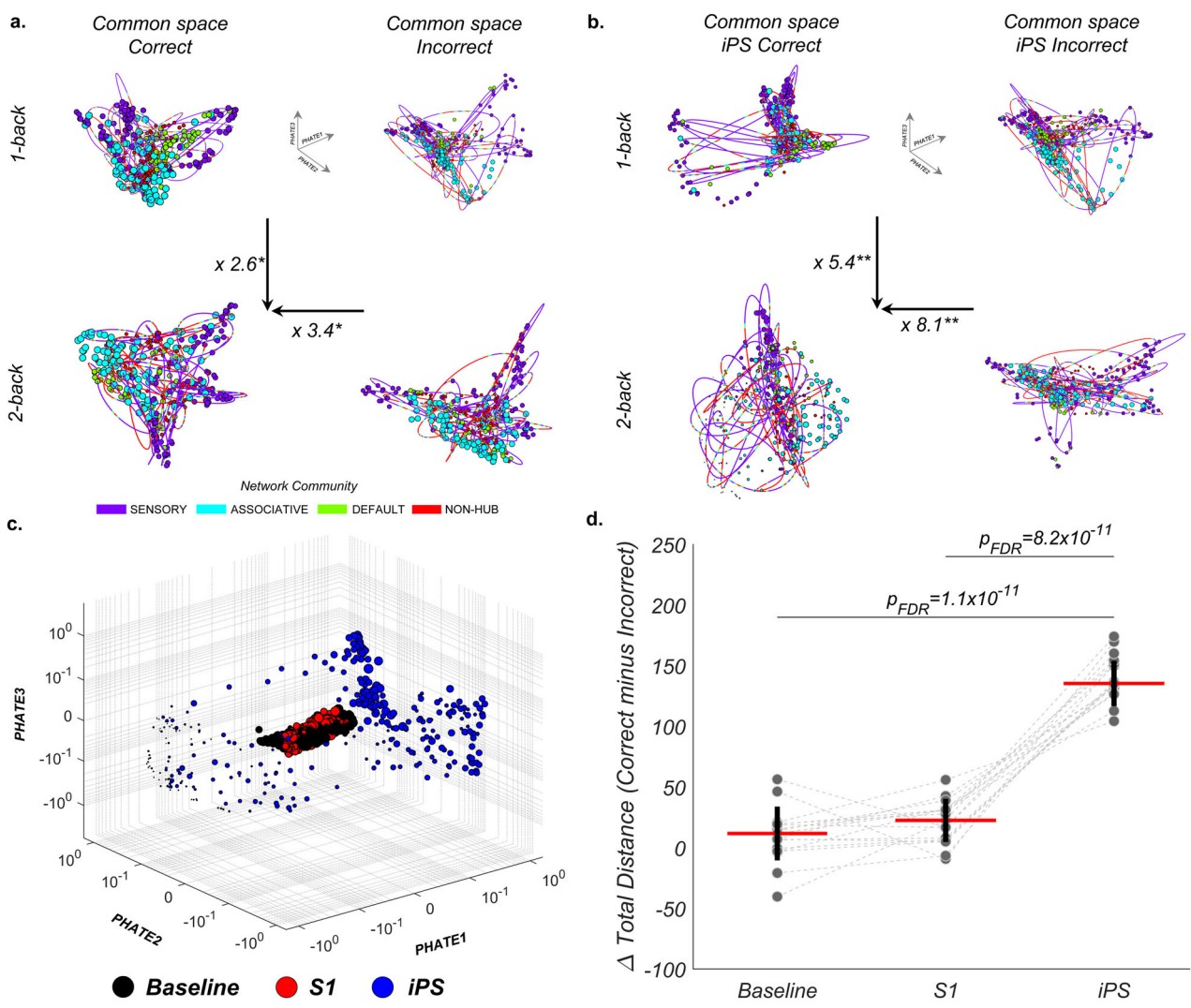

**Fig. 2 Reconfiguration of the low-dimensional trajectories of brain activity as a function of working memory load, task performances, and local neural perturbations. a** A shared (common) embedding space allows an across-session comparison of the state-space embedding of trajectories of brain activity supporting "correct" and "incorrect" trials. A general expansion of the trajectories of activity (i.e., utilizing data from all experimental sessions) is required to support "correct" trials in the 2-back condition (2.6 times greater than 1-back trials and 3.4 times greater than 'incorrect' trials in the same conditions). **b** Analysis of 'correct' and 'incorrect' trials for the iPS session showed that a significant expansion of low-dimensional trajectories of brain activity following a targeted neural perturbation (TMS on iPS; baseline and S1 shown in Supplementary Fig. 4a) supported successful 2-back performance (5.4 times greater than 1-back trials and 8.1 times greater than "incorrect" trials). For panels (**a** and **b**), asterisks indicate significant (*$p_{FDR}$ < 0.001) and highly significant values (**$p_{FDR}$ < 1 × 10$^{-6}$). **c** The significant expansion of trajectories following TMS on iPS involved the three PHATE dimensions ('correct' trials in the 2-back condition, visualized in log space, same embedding space across experimental sessions). **d** At the participant-level (n = 17, examined over three experimental sessions), comparison of the lengths of trajectories supporting distinct working memory loads (2-back minus 1-back, subtracting 'incorrect' trials; median represented by red line; within-session S.D. shown with vertical black lines) showed that "correct" trials following TMS on iPS were linked to a greater expansion of the low-dimensional trajectories as a function of load compared to baseline and S1 trajectories. Comparisons of trajectory lengths across sessions were tested with two-sided paired t-tests, with significance values corrected for multiple comparisons. Source data are provided as a Source data file.

$p = 2.8 \times 10^{-35}$; post-hoc paired t-test: iPS > baseline $p_{FDR} = 1.1 \times 10^{-11}$, iPS > S1 $p_{FDR} = 8.2 \times 10^{-11}$). Altogether, these findings suggest that a targeted perturbation of regional neural activity supporting task performance (iPS) caused a significant reshaping of the low-dimensional trajectories. These changes in low-dimensional trajectories are in line with the suggestion that whole-brain dynamics play a crucial role in maintaining behavioral stability following external perturbations[16].

**Association between changes in low-dimensional trajectories of brain activity and behavior.** Based on our previous work[1,4], we predicted that successful task performance should reflect an

optimal engagement of the low-dimensional trajectories of brain activity. Thus far, our results suggest that correct trials in the 2-back condition following a targeted perturbation of iPS activity rely on a marked expansion of the low-dimensional trajectory. This expansion was driven by an increased engagement of the first dimension (PHATE1) when compared to the same trials in the baseline and the S1 sessions (Supplementary Fig. 3a). Crucially, the increase in the group trajectory's length occurred in the absence of significant behavioral changes across sessions (within-subject ANOVAs, 1-back accuracy $p = 0.51$ and reaction time $p = 0.92$, 2-back accuracy $p = 0.59$ and reaction time $p = 0.68$) and was caused by a group-level suppression of task-evoked

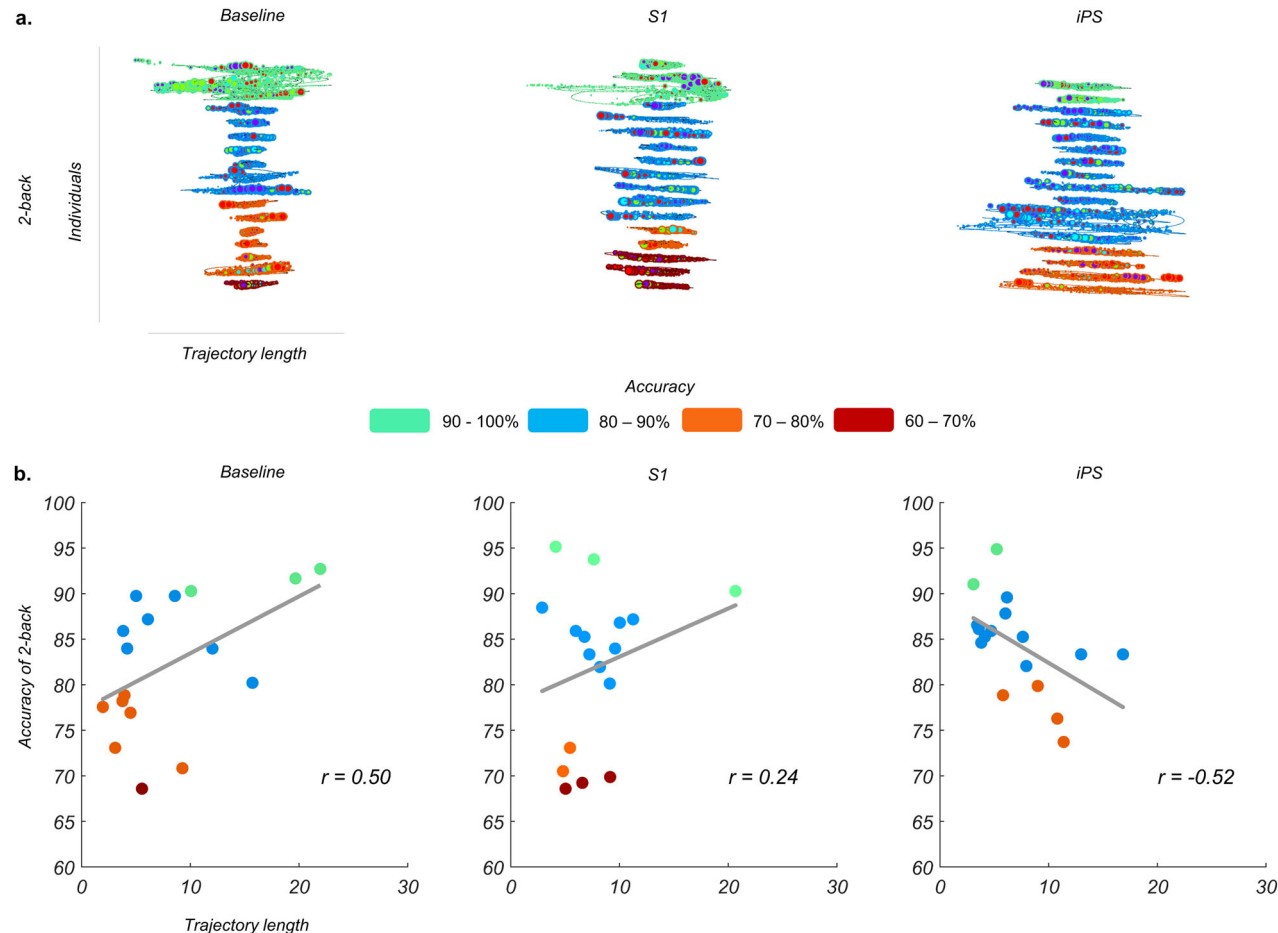

**Fig. 3 Associations between individual low-dimensional trajectories of brain activity and working memory performance. a** Analysis of the total length of low-dimensional trajectories for correct 2-back trials (measured across PHATE1, PHATE2, and PHATE3 dimensions) revealed a gradient linking individual trajectory length and behavioral accuracy across experimental sessions. Here each participant is tiered by overall task accuracy bands (% of correct 2-back trials). **b** Linear relationship between participants' trajectories and task accuracy for each session. In the baseline and the S1 session, longer trajectories link to higher 2-back accuracy whereas following perturbation of iPS, longer trajectories corresponded with lower task accuracy. Source data are provided as a Source data file.

neural activity in the iPS session compared to baseline (Supplementary Fig. 3c, d). An important unresolved question is to what extent the pronounced expansion of the trajectories of brain activity occupying the manifold reflects an adaptive response following local neural perturbation. To address this question, we evaluated the link between the length of each participant's correct trial trajectories and their task performance in the 2-back condition, across the three experimental sessions. We found opposite gradients linking the length of individual trajectories and performances in the baseline and the iPS sessions (Fig. 3a). In line with this observation, results showed a difference in the trajectories-behavior correlations between baseline and iPS ($p_{FDR} = 0.0051$). This difference was driven by a positive correlation at baseline ($r = 0.50$) and a negative correlation for the iPS session ($r = -0.52$, Fig. 3b). There was no significant difference in the correlations between baseline and S1 sessions ($p_{FDR} = 0.51$). An expansion of the low-dimensional trajectory of brain activity appears therefore necessary to support challenging working memory trials. However, preserving performance following targeted perturbation on a task-relevant brain region appears contingent on constraining the expansion of low-dimensional trajectories within a specific bandwidth, with excessive engagement hindering cognitive behavior.

By demonstrating a causal link between task-related activity in local neural circuits and low-dimensional brain dynamics, our findings advance knowledge on across-scale functional interactions supporting behavior. In the absence of significant behavioral changes across sessions, the reconfiguration of low-dimensional dynamics following neural stimulation of a task relevant region likely reflects the ability of the brain to accommodate acute perturbations. As the present work focuses on working memory related patterns of activity, the generalizability of our findings to other behavioral domains remains to be established. The current results also support intuitions from prior studies by showing that changes to cognitive capacity are linked to the functional cooperation of specialized brain regions and networks[17,18,19]. We note that the approach adopted herein did not assess changes in functional connectivity between task-relevant regions. Future studies are therefore required to evaluate the relationship between the defined low-dimensional trajectories of brain activity and measures of connectivity. Finally, our work motivates future clinical studies assessing how system-wide reconfigurations in low-dimensional brain dynamics may link to symptoms of disorders. These research endeavors may provide important information to orient the development of targeted brain stimulation therapies for mental disorders characterized by

complex deregulation of both local and brain-wide patterns of brain activity[20–24].

## Methods
We collected fMRI data in 17 participants (6 males and 11 females; age range between 18 and 35 years) to complete a working memory (*n*-back) task during three MRI sessions (totaling 51 experimental sessions): (i) a baseline session (before any brain stimulation), (ii) following TMS on iPS, and (iii) following TMS on S1[13]. The iPS and S1 sessions were counterbalanced across participants. Trial-locked hemodynamic responses from the *n*-back working memory task were then projected into a low-dimensional embedding space to characterize the spatiotemporal trajectories of fMRI signals as a function of working memory load. These trajectories were also used to examine within-subject changes in low-dimensional brain dynamics across the three fMRI sessions.

All participants provided informed consent and received financial compensation for their time ($20 per hour for each experimental session). The procedures were approved by the Committee for the Protection of Human Subjects at the University of California, Berkeley.

**Neuroimaging data acquisition**. fMRI data were acquired using a Siemens 3T Tim/Trio scanner at the Henry H. Wheeler, Jr. Brain Imaging Center at the University of California, Berkeley, equipped with a 32-channel head coil with a multiband echo-planar imaging sequence (acceleration factor = 4, TR = 1 s; TE = 33.2 ms; flip angle = 40°; voxel size: 2.5 mm$^3$ isotropic voxels with 52 axial slices). Structural MRI data were also acquired: TR = 2530 ms; TE = 1.64/3.5/5.36/7.22 ms; flip angle = 7°; field of view = 256 × 256, 176 sagittal slices, 1 mm$^3$ voxels; 2× GRAPPA acceleration. For the iPS and the S1 sessions, fMRI data were collected within 10 minutes of receiving TMS (cTBS).

**Working memory task (n-back)**. Participants performed an n-back task while in the scanner. The n-back task is a well-established paradigm to test working memory[25]. Across the three neuroimaging sessions, participants were presented a sequential set of pictures randomly selected from a set of 120 pictures of human faces and buildings. The key experimental manipulation involved the modulation of working memory load (2-back versus 1-back). For the 1-back trials, participants were asked to decide whether the picture displayed matched the picture presented in the previous trial, whereas for the 2-back task participants were asked to decide whether the picture displayed was a match to the picture presented two trials back. Each fMRI run comprised trials for each task condition (1- or 2-back). At the start of each run, participants viewed a fixation cross for 3 s, followed by counter-balanced n-back task blocks interleaved with a rest task block (fixation cross). Each n-back task block started with a 2-s initiation cue, followed by 13 trials. Each trial started with an image displayed at center of the screen for 0.5 s, followed by a randomly jittered intertrial fixation that lasted 1.5–10 s. Two to four repetitions of 1-back or 2-back trials were used within each task block, with presentation sequences randomized separately. For the first five participants, the task comprised runs of 155 s (two 60-s n-back blocks interleaved with a 25-s rest block and 7 s of a final fixation). The remaining 12 participants undertook slightly longer runs (236 s per run), with each run consisting of three 60-s task blocks interleaved with two 30-s rest blocks and a 10-s final fixation. Importantly, the total number of trials across the three sessions for all participants was identical (78 trials for each n-back condition).

**Transcranial magnetic stimulation**. A cTBS[11] stimulation paradigm was applied via a MagStim Super Rapid 2 stimulator using a figure eight double air film coil (70 mm diameter). Typically, cTBS induces a suppression of neural activity in the targeted brain region that outlasts the period of stimulation[11,26]. Prior to cTBS, each participant's TMS stimulation intensity was derived via electromyography. The stimulation intensity was defined as 80% of the active motor threshold, which was obtained by measuring the motor evoked potential (amplitude >50 uV for 5 out of 10 consecutive TMS pulses) at the first dorsal interosseous muscle. The cTBS paradigm comprised a patterned stimulation of triplet pulses (50 Hz, 20 ms) repeated every 200 ms. The whole stimulation totaled 600 pulses and lasted 40 s. Two TMS cortical targets were stimulated (in a counterbalanced fashion) in each of the 17 participants: the intraparietal sulcus (iPS) and the medial primary somatosensory cortex (S1). The iPS target was defined based on a task-evoked method (details in Hwang et al.[13]). TMS cortical targets were specified by selecting the peak coordinates for each participant within bilateral iPS to guide whether participants received right hemisphere (13 participants) or left hemisphere (4 participants) stimulation. Note that the effects of the target TMS site (right or left hemisphere) on the low-dimensional trajectories linked to the S1 and the iPS sessions did not differ.

**Neuroimaging data processing**. Preprocessing was done using FMRIPREP (v1[27]). Participant structural (T1) images were corrected for intensity nonuniformity, skull stripped, segmented, and normalized to the ICBM152 Nonlinear Asymmetric template version 2009c via nonlinear registration (ANTS v2.1.0). For fMRI data, images were motion-corrected via FSL's MCFLIRT routine and registered to T1

images using boundary-based registration. Following this step, functional data were spatially smoothed using a 4 mm full-width-at-half-maximum Gaussian Kernel, and a nuisance regression was performed (AFNI's 3dDeconvolve) to remove linear drifts, signals from six rigid-body motion parameters and their temporal derivatives. The averaged signal from white matter and ventricles were also removed. Motion confounds were minimized by removing all fMRI volumes (prior to regression analyses) that exceeded framewise displacement greater than 0.2 mm. Following these preprocessing steps, the mean time-series were extracted from 333 pre-defined cortical regions of interest using a common brain atlas[28]. The choice of this atlas was based on our previous work[1,4], which used the same atlas. Participant behavioral responses were used to extract the "correct" and the "incorrect" mean time-series for both 1-back and 2-back trials. Within each task run, stimuli onsets were convolved with a canonical hemodynamic response function (HRF; SPM12, spm_hrf.m function; TR = 1 s). The first 9 s following the onset of the stimuli were modeled. The estimated BOLD responses across trials were concatenated to define low-dimensional trajectories. Importantly, each event was convolved using a basis set that consists of the canonical HRF plus its partial derivatives with respect to delay and dispersion[29]. This informed basis set models for relatively small changes in the latency and duration of the fMRI signal response, offering both flexibility (while protecting from over-fitting) and efficiency[30]. We also performed a control analysis wherein low-dimensional trajectories across sessions and conditions were reconstructed from the stimuli onsets convolved with the canonical HRF only (i.e., without the informed basis set). Results showed that the inclusion of basis functions, such as the temporal derivative, are required to fully account for changes in low-dimensional dynamics across sessions and conditions. For each participant and experimental session (baseline, S1, and iPS), 1-back and 2-back time-series associated to 'correct' and 'incorrect' trials were extracted and concatenated. Across trials, conditions, and sessions, discontinuities or temporal sharp edges in the convolved stimuli time-series were evaluated using spectral kurtosis[31]. The stimuli time-series contained less than 7% of sharp edges or discontinuities (baseline 1.1%, S1 6.7%, iPS 5.8%). Within-session group time-series were also saved, with time-series from all participants also concatenated into 'correct' or 'incorrect' trial responses. We also combined the participants' within-session data to obtain time-series in a common embedding space. Finally, we estimated the magnitude of fMRI signal within the iPS brain region—for each participant across both baseline and iPS sessions by calculating the instantaneous signal power (analytic component of the Hilbert transform). iPS fMRI signals were extracted from a 10 mm (radius) sphere centered on a representative group median TMS coordinate (MNI in mm, x = 44, y = −51, z = 49).

**Low-dimensional trajectories of brain activity**. To project task-evoked fMRI responses into a low-dimensional embedding space, we employed a dimensionality-reduction method (Potential of Heat diffusion for Affinity-based Transition Embedding, PHATE[12]). Briefly, the PHATE method utilizes information geometry and manifold learning to derive local and global structures that exist within high-dimensional data. This approach generates a denoised and cleaned low-dimensional representation. Based on our previous work[1,4], we initially specified the dimensionality reduction of the mean stimuli time-series to five dimensions. To encode local information, the PHATE method first computed pairwise Euclidian distances across 333 brain regions[28] for each trial condition followed by the application of a kernel function (α-decay kernel = 35) to transform these distances into a normalized affinity matrix. Next, a diffusion process encodes global relationships present within the affinity matrix where the parameter *t* (Von Neumann entropy) ascribes random walk probabilities between one data point and the next. For this denoising step, we set $t_{mean} = 32$ ($t = 32 \pm 0.89$ S.D. across all participants, experimental sessions, and working memory conditions). An information-theoretic method, known as potential distance, is then used to measure the dissimilarity between random walk probabilities and compute an informational distance, where a nearest neighbor parameter ($k_{nn} = 10$, default value) ensures that the overall distance is not dominated by a specific cluster of data points. Lastly, a distance embedding (metric multidimensional scaling) squeezes the variability of high dimensional potential distances into a low-dimensional embedding space. Here the variability is maximally retained by the specified number of dimensions, providing an informative embedding space. For robustness testing we confirmed the estimation of low-dimensional trajectories with a range of α-decay kernel values between 35 and 50 (increments of 5) and a nearest neighbor $k_{nn}$ values between 6 and 10 (increments of 2).

To generate a whole-brain representation of low-dimensional dynamics we projected PHATE1, PHATE2, and PHATE3 values into the Gordon-333 brain parcellation[28]; for baseline, S1 and iPS sessions. Further, to facilitate the functional interpretation of low-dimensional dynamics we adopted a "connector hub" assignment[15]. Here, the 333 brain regions were clustered into four non-overlapping communities defined as the control-processing hub, cross-control hub, control-default hub, and a non-hub community. Regarding the three hub community assignments: (i) visual, auditory, and somatosensory brain regions constituted control-processing hubs; here defined as Sensory network community; (ii) cingulo-opercular, dorsal attention, frontoparietal, and salience regions constituted cross-control hubs; here defined as the Associative network community; and (iii) default mode and retrosplenial-temporal regions formed the control-default hubs; here defined as the Default network community. The non-hub network community

included ventral attention regions and the remaining brain parcels. Low-dimensional trajectories defined by the first three PHATE dimensions were plotted and visualized using this network community description, allowing us to appreciate brain regions engaged in state-space (Figs. 1, 2 and Supplementary Fig. 2).

To confirm the functional relevance of the inferred low-dimensional trajectories (Fig. 2a, b), we related the high-dimensional projections of the first three PHATE dimensions to spatial maps linked to NeuroSynth terms. NeuroSynth feature terms were derived from a set of 50 topics (https://neurosynth.org/analyses/topics/v5-topics-50/). Using a similar approach to Marguiles et al.[32], we selected the top 24 terms (z-statistic greater than 3.1). Terms with lower z-statistics were excluded. We first calculated a representative whole-brain mask (average across the three PHATE dimensions and experimental sessions) of correct responses from 2-back trials. Next, we calculated the functional correlates between regions in the representative PHATE mask and each NeuroSynth feature term. The resulting associations were binned from 0 to 100 in steps of 5%. Here, regions in higher percentile bands had greater association to the PHATE derived map. All feature terms were ordered according to the weighted mean for visualization purposes (Supplementary Fig. 2).

**Statistical analysis**. Within-subjects ANOVA and paired t-tests (two-sided) were used to compare the length of the low-dimensional trajectories as a function of working memory load (1-back versus 2-back) and performance ("correct" versus "incorrect" trials). To assess bivariate associations, the Pearson's correlation coefficient was used. The D1 method[33] was used to compare dependent correlations (length of low-dimensional trajectories with overall 2-back accuracy) across sessions. Multiple comparisons were corrected for by the Benjamini–Hochberg procedure to control false discovery rate (FDR).

**Reporting summary**. Further information on research design is available in the Nature Research Reporting Summary linked to this article.

## Data availability
All data generated and evaluated are available on the following GitHub repository: https://github.com/kneuro/trajectories_perturbations. The GitHub repository has the following https://doi.org/10.5281/zenodo.5568470. The repository contains all data required to reproduce the results. The raw and preprocessed fMRI data are available from the authors upon reasonable request. The web interface to the NeuroSynth database is available here: www.neurosynth.org. Source data are provided with this paper.

## Code availability
Custom MATLAB scripts used to analyze data are available at https://github.com/kneuro/trajectories_perturbations. The Github repository containing the python code used to query the NeuroSynth database, located at: https://github.com/neurosynth/neurosynth.

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

## Acknowledgements
K.H. and M.D. acknowledge the National Institutes of Health (NIH) (RO1 MH063901, F32 NS090757); and the National Science Foundation through their Major Research Instrumentation Program (BCS-0821855). L.C. is supported by the Australian National Health Medical Research Council (2001283 and 1138711). We thank Michael Breakspear and James Roberts for their advice on revisions to the manuscript.

## Author contributions
K.K.I., J.M.S. and L.C. designed the research; K.K.I., K.H., M.D. and L.C. contributed data and analytic tools; K.K.I. and L.C. analyzed the data; K.K.I. and L.C. wrote the paper; K.K.I., K.H., L.J.H., E.M., J.M.S., M.D. and L.C. provided feedback on the drafts of the paper. L.C. supervised the work.

## Competing interests
The authors declare no competing interests. L.C. is co-founder and co-director of a clinical neuromodulation centre (Qld. Neurostimulation Centre). This centre was not involved in the present work.

**Additional information**

