## [Peer Review File · Nature Communications]

Focal neural perturbations reshape low-dimensional trajectories of brain activity supporting cognitive performanceREVIEWER COMMENTS

Reviewer #1 (Remarks to the Author):

The authors present another piece of a continually evolving hypothesis around the expression of cognition through the navigation of low-dimensional "manifolds" that are formed by the network interactions across time. In the present paper, they identify a space traversed by a collection of networks related to the IPS and working memory (Cingulo-opercular, dorsal attention, and frontoparietal), whose geometry relates to working memory performance. Moreover, perturbing this space through TMS to the IPS changes the space, which impacts behaviour.

I appreciate the general message of the paper. I do fear, however, that the clarity of the message is somewhat clouded, due in part to the short format, which constrains some key information. I have 3 concerns

1) the terminology manifolds, trajectories, flows, and space seems to be used interchangeably. The way these are used in dynamical systems theory is distinct (e.g., Haken and Jirsa), and I would encourage the authors to abide by these definitions. What is estimated in this paper are trajectories that probably relate to an attractor. These may or may not be part of the same manifold. The manifolds, in the case of systems like the brain, occupy a relatively low-dimensional space. I think it's important to be consistent on these terms because there is growing confusion and often almost careless use of the terms by others that could threaten the scientific utility (c.f. functional connectivity).

2) I do not understand how the trajectories are reconstructed from the fMRI data. As I read the methods, the time series is convolved with a canonical HRF of 9 secs in SPM (basically 9 time points given the TR), which I think usually gives only a coefficient for the fit to the HRF unless temporal derivatives are estimated. Each epoch is then concatenated for a given trial type to get the trajectory. There are several concerns here:

- a) the temporal discontinuities between events make the actual and estimated trajectories hard to envision. How do the authors evaluate this?
- b) if there are differences in signal amplitude, I would imagine the space spanned by the first three components would differ between tasks only because of amplitude and not network affinity per se. The amplitudes would correlate with performance as the networks chosen are those that are typically related to working memory.

3) I appreciate the assertion that TMS provides some causal link between manifold architecture and behaviour. Philosophical issues aside with this argument, I would like to better understand whether the TMS effect is primary to the behaviour or is it a reflection of the reconfiguration of the space once a key node is perturbed? This is not an easy distinction and one that confounds most lesion work (permanent or temporary), but I think it is important to clarify this.

a) I may have missed it, but was there a behavioural effect of TMS on working memory performance?

signed
AR McIntosh

Reviewer #2 (Remarks to the Author):

This is a review of the manuscript "Focal neural perturbations reshape the low-dimensional brain activity flow supporting cognitive performance" by K.K. Iyer et al.

The authors evaluated brain network activity during performance of an N-back task with BOLD-fMRI, then applied theta-burst (inhibitory) TMS to a task-activated site, iPS, (compared to a non-task somatosensory site), and then analyzed brain network activity from task-fMRI immediately following stimulation. The authors calculated low-dimensional manifold from task-related, low-

frequency BOLD-fluctuations and evaluated the effect of TMS on this low-dimensional embedding space.

The study is motivated by a clear hypothesis and tested with appropriate methods and study design. The authors found that task-related perturbation led to largest manifold changes in the most demanding (2-back) WM-condition. This is supported by behavioral data that correct trials were related to a marked expansion of the manifold.

I have one question with respect to explained variance by PHATE 1-3. From Suppl Table 1 it seems that explained variance after iPS distributes more across PHATE 1 & 2, while PHATE 1 explains the majority in baseline and after S1. How could a shift towards two compared to one dimension be quantified and/or explain the findings in Fig. 1b or Fig. 2a/b?

Another question is related to behavioral performance. Some studies show an adaptation/learning effect of repeated N-back task performance. As I understand the design, the baseline always came first, but were the perturbation site sessions counter-balanced across subjects?

Overall, I found the study well performed and concisely presented.

Reviewer #1

The authors present another piece of a continually evolving hypothesis around the expression of cognition through the navigation of low-dimensional "manifolds" that are formed by the network interactions across time. In the present paper, they identify a space traversed by a collection of networks related to the IPS and working memory (Cingulo-opercular, dorsal attention, and frontoparietal), whose geometry relates to working memory performance. Moreover, perturbing this space through TMS to the IPS changes the space, which impacts behaviour.

I appreciate the general message of the paper. I do fear, however, that the clarity of the message is somewhat clouded, due in part to the short format, which constrains some key information. I have 3 concerns:

Authors (A): We appreciate the reviewer's positive appraisal of our work and for their important conceptual and technical suggestions, all of which have been incorporated in the revised manuscript.

R1.1. The terminology manifolds, trajectories, flows, and space seems to be used interchangeably. The way these are used in dynamical systems theory is distinct (e.g., Haken and Jirsa), and I would encourage the authors to abide by these definitions. What is estimated in this paper are trajectories that probably relate to an attractor. These may or may not be part of the same manifold. The manifolds, in the case of systems like the brain, occupy a relatively low-dimensional space. I think it's important to be consistent on these terms because there is growing confusion and often almost careless use of the terms by others that could threaten the scientific utility (c.f. functional connectivity).

A: We thank the reviewer for highlighting the importance of using appropriate terminology when describing concepts related to dynamical systems theory and their suggested references in improving our manuscript text. We agree that consistency and clarity around these terms are essential to provide a clearer motivation for the analyses and facilitate the interpretation of the results. Below we provide a clarification of the terminology and highlight the key revisions made in the manuscript:

Manifold: The term manifold implies that brain dynamics can be relatively low-dimensional and smooth. Brain dynamics linked to cognitive task performance evolve on a manifold in state space^{1,2}.

Velocity vector field in state-space: Mathematical objects that capture the dynamic properties of the system³, here the brain. Each point in state-space represents a possible state of the system. The velocity vector field describes the rate of change of the system's state variables at each point in state space^{1, 3, 4}. Thus, velocity vector fields define the rules of the system's behavior according to task specific constraints.

Low dimensional trajectory/ies: Reflects how the system's activity moves through state space according to the velocity vector field⁴.

In line with the above definitions, the following key changes have been made to the manuscript:

Title (p.1):

Focal neural perturbations reshape low-dimensional trajectories of brain activity supporting cognitive performance

Abstract (p.1):

Functional neuroimaging demonstrates that brain activity supporting adaptive behavior is constrained to low-dimensional manifolds. In human participants, we tested whether these low-dimensional constraints preserve working memory performance following local neuronal perturbations. We showed

that **specific** reconfigurations of **low-dimensional trajectories of brain activity** sustain effective cognitive performance following causal manipulation of local activity on, but not off, **the space traversed by these trajectories**. In this work we highlight a causal association between multi-scale **changes in brain activity** and cognitive function.

Introduction section (p.1, first and second paragraph):

Cognitive functions are supported by the activity of large-scale brain networks that adhere to a **relatively** low-dimensional manifold^{2, 5}. The study of low-dimensional brain dynamics allows us to capture whole-brain functional interactions between remote neural populations supporting behaviour⁶. Accordingly, recent functional neuroimaging work has shown that **trajectories of brain activity evolving on low-dimensional manifolds** scale with cognitive task complexity and are sensitive to performance errors⁷. These trajectories reflect how a system's activity changes according to the velocity vector field in state-space^{1, 3}. In essence, the vector field defines the rules of the system's behavior according to the constraints of a specific task and inform the evolution of trajectories of brain activity onto a low-dimensional sub-space (manifold)^{3, 4}. The term "manifold" implies that the system's dynamics can be relatively low-dimensional and smooth. Thus, changes in cognitive demands may alter the vector field and the trajectories of brain activity occupying a sub-space in state-space^{2, 4, 8, 9}.

Findings from **previous neuroimaging work** suggest that a reshaping of **the trajectories of brain activity evolving on low-dimensional manifolds** may capture key functional adaptations to offset perturbations of neural activity in specialized brain regions supporting cognitive functions^{5, 7}. If, and to what degree, the reshaping of **low-dimensional trajectories** is causally affected by local neural perturbations remains unknown. Addressing this question is critical to bridge the knowledge gap on the causal relation between local and **system-wide** brain activity patterns supporting cognitive functions¹⁰.

Additional edits have been made to the following sections:

Results:

- **p.2, first paragraph, third paragraph**
- **p.3, Fig. 1 legend**
- **p.4, first paragraph**
- **p.4, Fig. 2 legend**
- **p.4, Fig. 2c, adjusted to same axis limits across three PHATE dimensions**
- **p.5, first paragraph**
- **p.5, Fig. 3 legend**

Methods:

- **p.7, first paragraph**
- **p.8, second paragraph**

Supplementary Material

- **Supplementary Fig. 1 to 4, legends**

R1.2a: I do not understand how the trajectories are reconstructed from the fMRI data. As I read the methods, the time series is convolved with a canonical HRF of 9 secs in SPM (basically 9 time points given the TR), which I think usually gives only a coefficient for the fit to the HRF unless temporal derivatives are estimated. Each epoch is then concatenated for a given trial type to get the trajectory. There are several concerns here: a) the temporal discontinuities between events make the actual and estimated trajectories hard to envision. How do the authors evaluate this?

A: The reviewer makes an important point regarding the potential impact of temporal discontinuities for the reconstruction of the low-dimensional trajectories. This comment prompted us to clarify our methods and perform additional analyses on the data.

We now note that the potential temporal discontinuities between events were accounted for by including a mixture of basis functions that capture both the amplitude and the temporal characteristics of the hemodynamic response¹¹. To address this important comment, we have added the following text in the Results and Methods sections:

Results (p.2, first paragraph):

We estimated trial-locked fMRI responses by using a mixture of basis functions that capture both the amplitude and the temporal characteristics of the hemodynamic response (Methods).

Methods (p.8, first paragraph):

Importantly, each event was convolved using a basis set that consists of the canonical HRF plus its partial derivatives with respect to delay and dispersion¹². This informed basis set models for relatively small changes in the latency and duration of the fMRI signal response, offering both flexibility (while protecting from over-fitting) and efficiency¹¹.

In addition to the above clarification, we have also performed the following control analysis: We used spectral kurtosis¹³ to detect sharp temporal edges and discontinuities in the data that may have contaminated the definition of low-dimensional trajectories. As a representative example, we here report the results of the analysis conducted on group-level concatenated time series from correct 2-back trials. The mean percentage ($\% \pm$ standard error, S.E.) of sharp temporal edges or discontinuities detected between epochs across sessions was low: mean $4.5\% \pm 1.7\%$ S.E. (1.1% baseline, 6.7% S1, 5.8% iPS).

This result suggests that sharp edges or temporal discontinuities are unlikely to be a major concern for the original reconstruction of the low-dimensional trajectories. We note that the 4.5% value is markedly lower than values obtained using shorter or longer response windows which are more likely to contain discontinuities between concatenated events:

- $26.1\% \pm 5.8\%$ S.E. for 5 seconds windows after stimulus onset (windows do not capture the full stimulus response).
- $26.8\% \pm 17.1\%$ S.E. for 14 seconds windows after stimulus onset (response “contaminated” by the start of the next trial).

In taking the above into consideration, the following text has been added in the revised manuscript (Methods, p.8, first paragraph):

Across trials, conditions, and sessions, discontinuities or temporal sharp edges in the stimuli time-series were evaluated using spectral kurtosis¹³. Stimuli time-series contained less than 7% of sharp edges or discontinuities (baseline 1.1%, S1 6.7%, iPS 5.8%).

R1.2b: If there are differences in signal amplitude, I would imagine the space spanned by the first three components would differ between tasks only because of amplitude and not network affinity *per se*. The amplitudes would correlate with performance as the networks chosen are those that are typically related to working memory.

A: Differences in fMRI signal amplitude between events may indeed have an impact on the estimation of the low-dimensional trajectories of brain activity. However, it is improbable that the fMRI signal amplitude *per se* fully explains the detected changes in low-dimensional trajectories across sessions. To estimate the contribution of signal amplitude (alone) in explaining the results, we re-ran the analyses by convolving the fMRI signal with the canonical HRF only (i.e., without the informed basis set). The results showed:

- Within-session embedding spaces: We observed a *similar* degree of expansion in low-dimensional trajectories between 1-back and 2-back correct trials across all sessions (baseline 1.4x, S1 1.7x, and iPS 1.1x).

- **Shared embedding space (data collapsed across the three sessions):** The degree of expansion of trajectories between all 1-back to 2-back correct trials and all incorrect to correct 2-back trials *was the same* (1.2x, for both analyses).
- **Finally, the estimation of low-dimensional trajectories accounting only for changes in signal amplitude *did not* yield any significant correlations with individual performance** ($r_{baseline} = 0.46$, $r_{S1} = 0.19$, $r_{IPS} = -0.30$; all $p > 0.05$).

We note that sharp temporal edges and discontinuities between concatenated events used for these control analyses were similar (mean 5%) to those in time series modelled with the canonical HRF and a mixture of basis functions.

Finally, we also tested for differences in signal amplitude across the experimental sessions. As a representative example, we found no significant differences in the mean amplitude of concatenated time series from correct 2-back trials. Further, we also found no statistically significant differences in the mean amplitudes of specific network communities (Sensory, Associative, Default and Non-Hub).

Collectively, results from the above control analyses indicate that the detected effect cannot be simply explained by changes in signal amplitude alone. The following text has been added in the revised manuscript (Methods, p.8, first paragraph):

We also performed a control analysis wherein low-dimensional trajectories across sessions and conditions were reconstructed from the BOLD signal convolved with the canonical HRF only (i.e., without the informed basis set). Results showed that the inclusion of basis functions, such as the temporal derivative, are required to fully account for changes in low-dimensional dynamics across sessions and conditions.

R1.3a: I appreciate the assertion that TMS provides some causal link between manifold architecture and behaviour. Philosophical issues aside with this argument, I would like to better understand whether the TMS effect is primary to the behaviour or is it a reflection of the reconfiguration of the space once a key node is perturbed? This is not an easy distinction and one that confounds most lesion work (permanent or temporary), but I think it is important to clarify this.

A: We have now clarified that our results showed a significant reconfiguration of the group-level low-dimensional trajectories across experimental sessions, but *no* significant changes in behavior (see R1.3b below). This result, combined with the load-specific effects on trajectories and the across-subjects trajectory-behavior associations, suggest that TMS-induced changes in brain dynamics explain changes in behavior. Accordingly, the following text has been added in the Results section (p.5&6):

p.5, first paragraph:

Crucially, the increase in the group trajectory's length occurred in the absence of significant behavioral changes across sessions (within-subject ANOVAs, 1-back accuracy $p=0.51$ and reaction time $p=0.92$, 2-back accuracy $p=0.59$ and reaction time $p=0.68$) and was caused by a group-level suppression of task-evoked neural activity in the iPS session compared to baseline (**Supplementary Figure 3c&d**).

p.6, first paragraph:

Results also confirm intuitions from prior work by showing that changes to cognitive capacity are causally linked to the functional cooperation of specialized brain regions and networks^{14, 15, 16}. In the absence of significant behavioral changes across sessions, the reconfiguration of low-dimensional dynamics following neural perturbations on a task relevant region likely reflects the capacity of the brain to accommodate acute perturbations.

R1.3b: I may have missed it, but was there a behavioural effect of TMS on working memory performance?

A: Task accuracy and reaction time were similar across the three experimental sessions, for both working memory load conditions (1- and 2-back). We have now added the following text (in addition to the above):

Results (p.2, second paragraph):

As reported in our previous work¹⁷, these differences cannot be explained by group-level changes in behavioral performances (accuracy or reaction time) across the three experimental sessions (baseline, S1, and iPS).

Reviewer #2

The authors evaluated brain network activity during performance of an N-back task with BOLD-fMRI, then applied theta-burst (inhibitory) TMS to a task-activated site, iPS, (compared to a non-task somatosensory site), and then analyzed brain network activity from task-fMRI immediately following stimulation. The authors calculated low-dimensional manifold from task-related, low-frequency BOLD-fluctuations and evaluated the effect of TMS on this low-dimensional embedding space.

The study is motivated by a clear hypothesis and tested with appropriate methods and study design. The authors found that task-related perturbation led to largest manifold changes in the most demanding (2-back) WM-condition. This is supported by behavioral data that correct trials were related to a marked expansion of the manifold.

Overall, I found the study well performed and concisely presented.

Authors (A): We appreciate the reviewer's positive appraisal of our work and the constructive feedback to improve the manuscript. The comments have been carefully considered and fully addressed below.

R2.1: I have one question with respect to explained variance by PHATE 1-3. From Suppl Table 1 it seems that explained variance after iPS distributes more across PHATE 1 & 2, while PHATE 1 explains the majority in baseline and after S1. How could a shift towards two compared to one dimension be quantified and/or explain the findings in Fig. 1b or Fig. 2a/b?

A: The reviewer rightly points out that the first two dimensions (PHATE1 and PHATE2) explain the bulk of the signal variance in the iPS session while the majority of the variance is captured by the first dimension (PHATE1) in the baseline and S1 sessions. Supplementary Table 1 shows that, overall, PHATE1 accounts for most of the brain activity: 83% on average for baseline and S1; 72% on average for iPS. This result suggests that PHATE1 plays a major role in supporting task-relevant brain dynamics. The increased engagement of PHATE2 following TMS of iPS is linked to the observed low-dimensional expansion in this condition (Fig.2). That is, the shift in task-related variance from PHATE1 towards PHATE2 likely captures the brain's adaptive response to a targeted perturbation of activity of a task-relevant region (iPS).

We note that the axes of Fig. 2c have been adjusted to have the same axis limits. This small change allows a more direct appreciation of the variance across the three PHATE dimensions.

We have added the following text to further describe how shifts in low-dimensional signals were quantified and how these changes can be explained:

Results (p.2), second paragraph:

The variance captured by PHATE dimensions identifies the dominant low-dimensional signal in state-space, reflecting the global dynamics of brain activity present across experimental sessions and task

conditions. Our findings suggest that PHATE1 plays a key role in supporting task-related low-dimensional activity (83% on average for baseline and S1; 72% on average for iPS). The detected shift in task-related variance towards PHATE2 following perturbation of iPS is likely to reflect an adaptive response in the system's brain dynamics. As reported in our previous work¹⁷, these differences cannot be explained by group-level changes in behavioral performances (accuracy or reaction time) across the three experimental sessions (baseline, S1, and iPS).

R2.2: Another question is related to behavioral performance. Some studies show an adaptation/learning effect of repeated N-back task performance. As I understand the design, the baseline always came first, but were the perturbation site sessions counter-balanced across subjects?

A: As reported in our previous work (Hwang et al.¹⁷), we found no differences in accuracy and reaction time or evidence that adaptation/learning effects of N-back task performance had occurred across experimental sessions. This information has now been added to our manuscript text (previous response and response to R1.3b). In addition, we have now clarified that perturbation sites were counterbalanced across subjects:

Introduction (p.1, third paragraph):

The iPS and the S1 TMS sessions were counterbalanced across participants.

Methods (p.7, first paragraph, original text below):

The iPS and S1 sessions were counterbalanced across participants.

References

1. McIntosh, A.R. & Jirsa, V.K. The hidden repertoire of brain dynamics and dysfunction. *Network Neuroscience* **3**, 994-1008 (2019).
2. Ebitz, R.B. & Hayden, B.Y. The population doctrine revolution in cognitive neurophysiology. *Neuron In press* (2021).
3. Huys, R., Perdikis, D. & Jirsa, V.K. Functional architectures and structured flows on manifolds: A dynamical framework for motor behavior. *Psychological review* **121**, 302 (2014).
4. Pillai, A.S. & Jirsa, V.K. Symmetry breaking in space-time hierarchies shapes brain dynamics and behavior. *Neuron* **94**, 1010-1026 (2017).
5. Shine, J.M. *et al.* Human cognition involves the dynamic integration of neural activity and neuromodulatory systems. *Nature neuroscience* **22**, 289-296 (2019).
6. Breakspear, M. Dynamic models of large-scale brain activity. *Nature neuroscience* **20**, 340-352 (2017).
7. Shine, J.M. *et al.* The low-dimensional neural architecture of cognitive complexity is related to activity in medial thalamic nuclei. *Neuron* **104**, 849-855. e843 (2019).
8. Wong, K.-F. & Wang, X.-J. A recurrent network mechanism of time integration in perceptual decisions. *Journal of Neuroscience* **26**, 1314-1328 (2006).
9. Murray, J.D. *et al.* Stable population coding for working memory coexists with heterogeneous neural dynamics in prefrontal cortex. *Proceedings of the National Academy of Sciences* **114**, 394-399 (2017).
10. Park, B.-y. *et al.* An expanding manifold in transmodal regions characterizes adolescent reconfiguration of structural connectome organization. *Elife* **10**, e64694 (2021).
11. Friston, K.J., Penny, W. & David, O. Modeling brain responses. *Int Rev Neurobiol* **66**, 89-124 (2005).
12. Friston, K.J. *et al.* Event-related fMRI: characterizing differential responses. *Neuroimage* **7**, 30-40 (1998).
13. Vrabie, V., Granjon, P. & Serviere, C. Spectral kurtosis: from definition to application. 6th IEEE international workshop on Nonlinear Signal and Image Processing (NSIP 2003); 2003; 2003. p. xx.
14. Shine, J.M. *et al.* The dynamics of functional brain networks: integrated network states during cognitive task performance. *Neuron* **92**, 544-554 (2016).
15. Cohen, J.R. & D'Esposito, M. The segregation and integration of distinct brain networks and their relationship to cognition. *Journal of Neuroscience* **36**, 12083-12094 (2016).
16. Cocchi, L., Zalesky, A., Fornito, A. & Mattingley, J.B. Dynamic cooperation and competition between brain systems during cognitive control. *Trends in cognitive sciences* **17**, 493-501 (2013).
17. Hwang, K., Shine, J.M., Cellier, D. & D'Esposito, M. The Human Intraparietal Sulcus Modulates Task-Evoked Functional Connectivity. *Cerebral Cortex* **30**, 875-887 (2020).

REVIEWERS' COMMENTS

Reviewer #1 (Remarks to the Author):

I appreciate the extremely thorough response to my initial review. The additional analyses and clear terminology on manifolds have addressed my concerns. I believe this paper is ready for publication. I congratulate the authors on their superb work.

AR McIntosh

Reviewer #2 (Remarks to the Author):

Thank you, the authors have addressed my concerns in detail.